# Programmable material via thiol-ene polymerization initiated by electric-field induced thiyl radical on piezoelectric ZnO

Jun Wang[1,6], Zhao Wang [2,6], Jorge Ayarza[1], Ian Frankel[3], Chao-Wei Huang[1], Kai Qian [3], Yixiao Dong[1], Pin-Ruei Huang[1], Katie Kloska[1], Chao Zhang[1], Siqi Zou [1], Matthew Mason [4], Chong Liu [1], Nicholas Boechler [3,5] & Aaron P. Esser-Kahn [1] ✉

The spatial and temporal control of material properties at a distance has yielded many unique innovations including photo-patterning, 3D-printing, and architected material design. To date, most of these innovations have relied on light, heat, sound, or electric current as stimuli for controlling the material properties. Here, we demonstrate that an electric field can induce chemical reactions and subsequent polymerization in composites via piezoelectrically-mediated transduction. The response to an electric field rather than through direct contact with an electrode is mediated by a nanoparticle transducer, i.e., piezoelectric ZnO, which mediates reactions between thiol and alkene monomers, resulting in tunable moduli as a function of voltage, time, and the frequency of the applied AC power. The reactivity of the mixture and the modulus of a naïve material containing these elements can be programmed based on the distribution of the electric field strength. This programmability results in multi-stiffness gels. Additionally, the system can be adjusted for the formation of an electro-adhesive. This simple and generalizable design opens avenues for facile application in adaptive damping and variable-rigidity materials, adhesive, soft robotics, and potentially tissue engineering.

Controlling material properties with external forces has been transformative[1]. Light, heat, pH, mechanical force, and electric current have all been used to great effect to alter the composition and structure of materials[2,3]. The resulting materials can be patterned and printed into structures with control over length-scale, geometry, and stiffness[4]. Materials with programmable compliant properties are able to respond on-demand to environmental changes[5,6]. However, among all stimuli, an electric field uniquely excels at control, sensing, and distribution of power[7,8]. Several examples in the literature show that an electric field can induce physical changes in a material, such as phase transitions, which subsequently result in a change in shape or stiffness[9]. These materials are often referred to as electro-programmable materials[8]. For example, Silberstein et al. reported a comprehensive computational study on the stiffening of polyelectrolytes under an electric field[10]. The study showed that the elastic modulus of a polyelectrolyte can increase by 45% under a simulated electric field ($E = 0.2$ V/m) due to better chain alignment and an increase in the number of ionic crosslinks. However, the stiffening only occurs with charged polymers while under constant application of an electric field, meaning the material would soften once the electric field is removed[10]. In contrast, electrochemistry has been used to

[1]Pritzker School of Molecular Engineering, University of Chicago, Chicago, IL, USA. [2]College of Chemistry, Chemical Engineering and Materials Science, Soochow University, Suzhou, China. [3]Department of Mechanical and Aerospace Engineering, University of California San Diego, La Jolla, CA, USA. [4]Department of Chemistry, Princeton University, Princeton, NJ, USA. [5]Program in Materials Science and Engineering, University of California San Diego, La Jolla, USA. [6]These authors contributed equally: Jun Wang, Zhao Wang. ✉e-mail: aesserkahn@uchicago.edu

produce chemical reactions within materials, resulting in permanent structural modifications[1,8]. However, electrochemical reactivity is highly dependent on both the surface of the electrodes as well as the conductivity of the material[11]. Chemical reactions that can be triggered and controlled through the use of electric fields with minimal current flow thus offer an interesting alternative. Nevertheless, it has been difficult to use electric fields to directly induce chemical reactions in a material—relying only on surface electrodes or infeasibly high electric fields ($10^7$–$10^9$ V/m) to mediate chemical changes.

Previously, our group demonstrated a self-strengthening behavior in an adaptive material induced by applied vibration mediated by ZnO (piezoelectric) nanoparticles[12,13]. In observing the literature on piezoelectric energy-harvesting materials, we formulated a hypothesis that ZnO nanoparticles could be activated via an external electric field instead of mechanical vibration, as both electric-field-induced deformations and strain-induced internal electrical fields have been detected in ZnO nanostructures[14–16]. In this work, we present an alternative approach to conduct thiol−ene polymerization and crosslinking reactions via an applied electric field. This electrically-controlled reaction relies on low-strength electric fields (250–1000 V/m) to activate ZnO nanoparticles, which in turn promote the piezochemical generation of radicals and subsequent radical-mediated thiol−ene reaction. Additionally, this approach enables far-range chemical reactivity within a material, which permanently alters its composition and mechanical properties. We showcase two examples of the capabilities of this approach. In the first example, we placed an organo-gel embedded with multi-functional thiol−ene monomers and ZnO nanoparticles within an electric field and modified its stiffness by triggering a crosslinking reaction. (Fig. 1a). The elastic modulus of the resulting gel can be controlled by adjusting the parameters of the electric input (Supplementary Fig. 1). In a second example, we adjust the formulation and reaction conditions to produce a fast, electrically curable adhesive with strong binding properties to various surfaces.

## Results and discussion
### Electrically-controlled thiol−ene polymerization
First, we tested whether ZnO could initiate a polymerization when subjected to an electric field. We used the thiol−ene linear

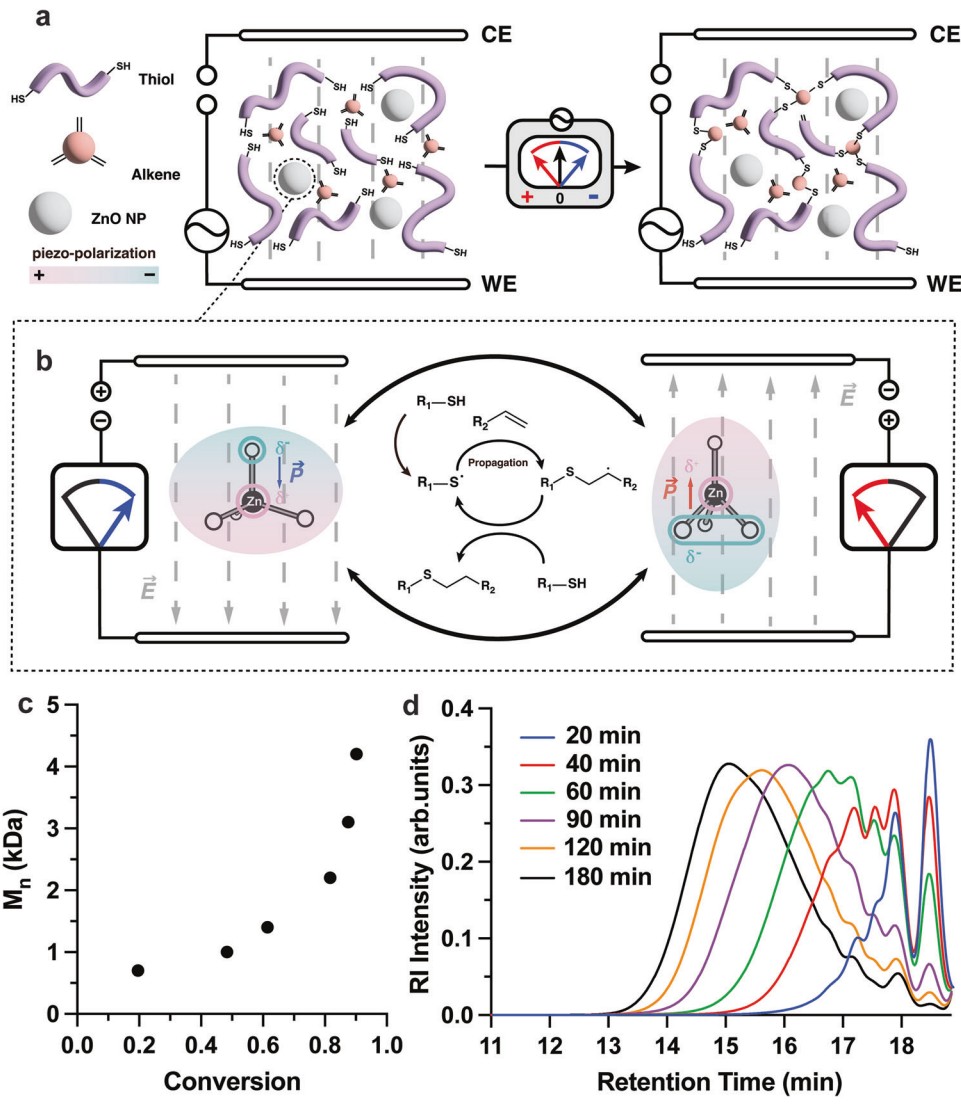

**Fig. 1 | Conceptual illustration of electric field-triggered thiol−ene reaction mediated by ZnO nanoparticles. a** Conceptual illustration of the thiol−ene reaction under an alternating electric field. **b** Piezo-polarization state of ZnO nanoparticles under electric field and plausible mechanism for the E-thiol−ene reaction. **c** Evolution of $M_n$ over increasing monomer conversion of polymer via E-thiol−ene polymerization. **d** GPC trace depicting the evolution of polymer molecular weight over time. The polymerization mixture contained 3.00 mmol of TEGDE, 3.00 mmol of EDT, 0.15 mmol $Et_4NBF_4$ in 1.50 mL DMF and 210 mg ZnO nanoparticles. The mixture was treated under 8 $V_{rms}$ and 500 Hz using the AC power supply at varied durations, denoted by the different line colors defined in the legend. Aliquots from the reaction were analyzed by [1]H-NMR and by GPC with polystyrene standards.

polymerization between tri(ethylene glycol) divinyl ether (TEGDE) and 2,2′-(ethylenedioxy) diethanethiol (EDT) as a model reaction. We chose tetraethylammonium tetrafluoroborate ($Et_4NBF_4$) as supporting electrolyte in DMF (Supplementary Fig. 2)[12]. A reaction mixture containing TEGDE (3 mmol), EDT (3 mmol), and piezoelectric ZnO nanoparticles (7.5 wt%) was subjected to an alternating electric field (8 $V_{rms}$, 500 Hz, 500 V/m). Experiments were conducted in the dark to eliminate the possibility of photocatalysis. The reaction progress was monitored by [1]H-NMR and gel permeation chromatography (GPC) at 1 h intervals. A short polymer (number-averaged molecular mass, $M_n$ = 4200 Da) was obtained with 90 % monomer conversion, showing the typical kinetic profile of a step-growth polymerization (Fig. 1c)[12,17]. In the GPC traces of Fig. 1d, the presence of the refractive index (RI) peak at longer retention times denotes that less reaction has occurred.

A plausible mechanism for the electric field-induced thiol−ene reaction is described in Fig. 1b. ZnO nanoparticles serve as an inverse piezo-electrochemical mediator to induce the formation of thiyl radicals for a thiol−ene reaction[18,19]. Our previous work with mechanically promoted polymerizations through the piezochemical effect had shown the unique property of piezoelectric, specifically ZnO, nanoparticles to promote the thiol−ene reaction through a combination of both surface chemistry and piezochemical activation. To confirm that the current reaction setup occurred through a comparable mechanism, albeit using an E-field as stimuli, we performed the control experiments without ZnO nanoparticles, or even replacing the ZnO with several different types of piezoelectric (PZT, $BaTiO_3$ and $BiFeO_3$) or photoelectric ($TiO_2$) nanoparticles. In all cases, no polymerization was detected, indicating that the presence of ZnO nanoparticles was essential for reactivity (Supplementary Fig. 4, Supplementary Table 1)[12,20]. We hypothesized that since the reactivity originated from the surface of nanoparticles, the interaction between thiol and ZnO would be essential for this chemistry to proceed.

To confirm that thiol adsorption onto ZnO nanoparticle surface is an essential feature of piezo-mediated reactivity, we compared XPS samples of different nanoparticles (ZnO/$BaTiO_3$) mixed with an EDT solution (2.0 M in DMF). The nanoparticles were first stirred with EDT solution for 3 h and then thoroughly washed with ethanol to remove unattached EDT molecules. The nanoparticles were eventually dried and characterized by XPS. (Supplementary Fig. 5). Compared to pure ZnO, in ZnO mixed with EDT, both the Zn 2$p$ and Zn 3$d$ peaks exhibited a noticeable shift towards lower binding energy by ~0.7 eV. This result matches a previous study of thiol adsorption on Zn, showing a nearly identical peak shift towards lower binding energy[21,22]. Also, the presence of a strong S 2$p$ peak of thiol-treated ZnO in the XPS indicates that the proton was dissociated from the thiol and that EDT bonded to a Zn site at the surface of ZnO. For $BaTiO_3$, we observed no such shifts. Further characterization of zeta potential also confirmed our hypothesis, wherein we measured the zeta potential under various electric fields under direct current (from $2 \times 10^3$ to $2 \times 10^4$ V/m) since negative surface charges of ZnO could be responsible for our previously reported mechanically mediated thiol−ene reaction (Supplementary Table 2). The results suggested that the change of the electric field led to very little change in surface charges. We also tested piezoelectric, silane-coated ZnO nanoparticles (1 wt%, ~0.2 nm coating) with a similar zeta potential of −22.6 mV. At the same concentration and electric field, no reactivity was observed (Supplementary Fig. 3)[12,23]. The above results provide strong evidence that the binding of EDT directly to the surface of ZnO promotes the energy transfer from ZnO to the S and provides a piece of the mechanism for privileged electric-field mediated reactivity of ZnO.

To further explore how the electric field affects the ZnO, cyclic voltammetry was conducted to evaluate the possibility of any redox transformation involving Zn. A soluble zinc salt in DMF, $Zn(BF_4)_2$, was also measured for comparison. No redox peaks were observed within the range from −1.2 to 1.0 V before and after 3 h of applying the electric field (Supplementary Fig. 6, blue and red lines). The CV curve of $Zn(BF_4)_2$ showed a redox peak of $Zn/Zn^{2+}$ reaction at −0.49 V; however, that peak is absent in the CV curves of the thiol−ene mixture, thus indicating there is no soluble $Zn^{2+}$ present in it, either before or after the application of the electric field.

While a complete mechanistic study will be the focus of future reports, we performed several preliminary control experiments to answer the most outstanding questions. The first question was whether the ZnO generated a radical equivalent during transduction. To test this, we added 4-methoxyphenol (MEHQ) as a free radical scavenger up to 0.60 mM and observed the inhibition of polythioether formation, indicating that the polymerization was mediated by a radical transfer process (Supplementary Table 3)[24]. A second question was whether an AC electric field was necessary to activate ZnO; therefore, we switched the power from AC to DC voltage (8 V, 500 V/m) for 3 h to explore how the reaction responded when only experiencing a constant electric field. Under DC, we observed lower reactivity, and after 3 h, the reaction ceased coincident with a small amount of reactive material fouling the bulk electrode surface (Supplementary Fig. 7). This result indicated that indeed a DC induction worked, but resulted in possible side reactions. A final question was whether the process was driven by either electron transport (current) or the electric field. To avoid direct electrical contact between the electrodes and the reaction mixture, we placed the Pt electrodes outside the plastic reaction cell. We conducted the same initial linear thiol−ene reaction in a black Faraday cage. The GPC results from Supplementary Fig. 8 confirm the formation of an oligomeric species with a $M_n$ of 3200 Da in the presence of the E-field without direct electrical contact. This adds evidence that current or electrohemistry is not required to mediate a reaction. The summary of the evidence indicates that this process is predominantly mediated by the electric field.

We have also observed that the mechanical activation of ZnO can occur via mechanical stirring (500 rpm) (see Supplementary Fig. 7). However, the mechanical activation of piezo-particles from stirring does not achieve nearly the same reactivity as the electric field. In the current approach, we hypothesize that a potential mechanism is that the external AC electric field induces the deformation of ZnO nanoparticles following the mechanism of the converse piezoelectric effect, wherein the nanoparticles expand and contract[25,26]. In this way, the strains in ZnO nanoparticles may modulate the local electric field, promoting the generation of thiyl radicals. This proposed mechanism is similar to that of the electrostatic catalysis studied by Coote et al., where properly oriented electric fields can catalyze originally improbable Diels−Alder reactions through stabilization of the resonance structures of transition states. In our case, the orientation of the electric field was fixed by the direct contact of thiols with the ZnO surface and the local piezoelectrically generated electric field at the ZnO surface, all of which resembles the scanning tunneling microscopy (STM) setup used by Coote et al. [27].

To determine the role atmospheric oxygen played in the E-thiol−ene reaction, we conducted additional experiments in a glovebox environment. The chemical kinetics data indicate that oxygen does not play a major role in the polymerization mechanism since we observed comparable conversions and molecular weights when the reaction was carried out in a glove-box versus exposed to the air (Supplementary Fig. 9). We conclude, therefore, that the necessary radicals for the propagation are produced from piezochemical reactivity between ZnO and the thiol monomer.

In addition, we explored how different exposure times to the electric field affected the polymerization kinetics. Kinetics data from Supplementary Fig. 10 shows that the reaction proceeds only when the electric field is applied for over 5 min. Additionally, we found that the overall conversion percentage increases with increased length or

power of the electric field input. These results show that the generation of radicals is positively correlated with the application of an electric field. Nevertheless, we observe a minimum amount of time that the electric field must be applied to generate enough radicals to trigger the polymerization. Additionally, there appears to be a process by which radicals are either consumed or inactivated for further reactivity, since we obtain higher conversions when the electric field is applied for longer. Critically, our E-thiol−ene polymerization kinetics is similar to those of thiol−ene, which are initiated by light—indicating that this process is of similar efficiency and rate as light-based approaches of similar power[28–30].

With this further understanding of the reaction kinetics and potential mechanism, we were eager to explore other potential reactions. It seemed possible that both acrylate polymerization and other thiol-mediated polymerizations might be amenable to ZnO polymerization. Based on the same mechanism, this chemistry can also be expanded to make acrylate homo-polymers. In this case, there is a potential mechanism in which thiyl radicals generated by E-field-induced piezo-chemistry can initiate the chain polymerization of methyl methacrylate monomers (Supplementary Fig. 11). However, low levels of polymerization were observed for disulfide-based monomers (Supplementary Figs. 12 and 13). We hypothesize that interactions with the ZnO surface may play an as-yet fully understood role in this process, which allows free thiols to be more reactive than disulfides[31].

## Electrically controlled thiol−ene gelation

Having demonstrated AC current-induced thiol−ene step-growth polymerization, we further sought a method to modify material properties via polymer crosslinking to form a gel. As such, we tested if the polymerization could create a crosslinked network of TTT and EDT when applying an AC electric field (500 Hz, 8 $V_{rms}$, 3 h) (Supplementary Figs. 2 and 14). The formed viscoelastic gel sample displayed a storage modulus of 776 kPa at 1 Hz (Supplementary Fig. 15). Of note, when a DC voltage was applied, an anomalous porous gel structure appeared, which was not seen under AC voltage (Supplementary Fig. 16). It became clear from these initial experiments that regulating the electric field including parameters such as duration, strength, and frequency influenced the gelation process and thus the generation of thiyl radicals.

First, to determine how the electric field strength would affect gel stiffness, we fixed the distance between the electrodes to 16 mm and changed the AC voltage from 4 to 16 $V_{rms}$ (500 Hz, 250–1000 V/m). The gelation onset time decreased and the storage modulus of the resulting gel increased monotonically from 346 to 776 kPa as the applied voltage increased (Fig. 2a and b).

To quantitatively characterize the thiol−ene gelation, we considered how the crosslinking reaction might proceed via a radical-mediated process and shared the similarity of radical generation of thiol−ene linear polymerization between TEGDE and EDT. To evaluate the mechanism, we measured the monomer conversion during thiol−ene linear polymerization in the presence of MEHQ (0.30 mM) while varying the applied voltage. As determined by $^{1}$H-NMR in Supplementary Table 4, a much higher level of conversion was achieved at 16 $V_{rms}$ (77.9%) than at 4 $V_{rms}$ (7.08%), indicating that increasing the voltage generated more radicals in the same amount of time. Given the exothermic nature of the addition between thiols and terminal alkenes[32,33], we further tracked the time-dependent temperature evolution of thiol−ene gelation at various applied voltages via a temperature sensor. The trend depicted in Supplementary Fig. 17 reveals

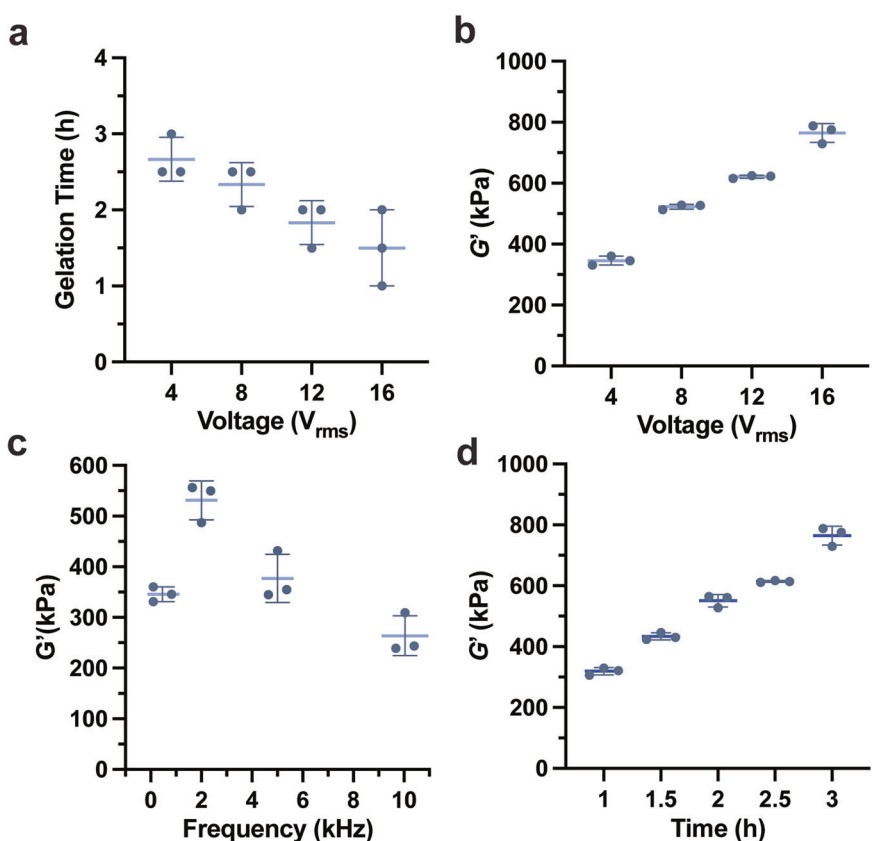

**Fig. 2 | Controllable modulus of the organo-gel to different electric inputs.**
**a** Gelation time as a function of AC voltage. Storage modulus ($G'$) of the gels after electric field application as a function of: **b** AC voltage intensity (500 Hz, 3 h), **c** frequency (4 $V_{rms}$, 3 h), and **d** reaction time (16 $V_{rms}$, 500 Hz). All the error bars are standard deviation calculated with the results of three independent experiments.

that more heat was released with increasing voltage, which suggests additional reactions occurring as a result of the higher electric field[34]. We also evaluated the temperature change of thiol−ene gelation with 0.60 and 1.20 mM MEHQ while applying the same voltage (16 $V_{rms}$, 500 Hz). There was no peak of the temperature profile found in the sample with 1.20 mM MEHQ (Supplementary Fig. 17B) due to the effective quenching of thiyl radicals generated from the electric field. In the other samples, we observed a peak in temperature followed by a plateau, which is indicative of an exothermic reaction. This experiment also confirmed that the change in temperature is not due to resistive heating from the application of the electric field. A thermally driven process would result in a linear increase in temperature as the reaction progresses.

We also sought to determine if changing the frequency of the AC current would affect reactivity. As such, we varied the frequency from 500 Hz to 10 kHz but kept the voltage and time constant (4 $V_{rms}$, 250 V/m, 3 h), and measured the resulting storage modulus ($G'$) of the gel. The resulting gels had a peak $G'$ at a frequency of 2 kHz, and the modulus varied with the frequency of the electric field (Fig. 2c). This may be due to larger deformations when the structural resonance of the test samples coincides with the driving frequency. Previous literature on ZnO micro- and nanoparticles also showed a decreasing trend in both effective dielectric permittivity $\varepsilon_r$ and effective piezoelectric coefficient $d_{33}$ while the frequency increases[26,35].

To further explore the effect of the electric field on the organogel, we used a laser Doppler vibrometer to measure the vibrational velocity of two thiol−ene organogels: one loaded with ZnO (piezoelectric semiconductor) and another one loaded with $TiO_2$ (non-piezoelectric semiconductor) to serve as a control. The results (Supplementary Fig. 18) showed no significant difference between the displacement of both samples. In the case of ZnO organogel, we theorize that the contribution to the displacement coming from the nanoparticles may be overshadowed by the displacement of the polymer matrix itself due to the dielectric elastomer effect or electrostriction. Additionally, both samples may have different structural resonances, which means comparing excitation amplitudes between the two structures may lose meaning depending on how close the excitation frequency of 2 kHz is to its own natural frequency.

To explore how gelation would vary with time, we evaluated the gels' storage moduli in the gelation time window. The samples were exposed to a higher AC electric field (16 $V_{rms}$, 500 Hz, 1000 V/m), and we tested their modulus every 30 min. As expected, as the field was applied, the longer the reaction proceeded, the more crosslinking occurred and a higher resulting storage modulus (Fig. 2d) was observed. The stiffness of the gel progressively increases throughout the gelation time window, ranging from 330 to 764 kPa in modulus. This trend indicates that such a method could be applied to a kinetically controlled crosslinking network formation.

We also explored how various ZnO concentrations affect the thiol−ene crosslinking reaction. First, we evaluated the gels' storage moduli under electric field curing (16 $V_{rms}$, 500 Hz, 3 h). As expected, the modulus of the resulting organogels increased steadily with ZnO concentration, suggesting that ZnO enhances radical generation through the piezoelectric effect, thus promoting efficient crosslinking (Supplementary Fig. 25). To rule out the possibility of ZnO acting as a mechanical filler, we subjected the thiol−ene mixture with varying ZnO concentrations to heat curing (100 °C, 24 h) until the organogel got fully crosslinked. Data from Supplementary Fig. 26 showed that the modulus plateaus beyond 1.5 wt% ZnO, indicating limited mechanical enhancement from additional ZnO usage. These findings confirm that ZnO's primary role is to facilitate crosslinking under electric fields through radical generation rather than serving as a filler material.

We conclude from the above experiments that voltage, reaction time, frequency (potentially via its interplay with the sample structural resonance), and ZnO concentration all play critical roles in the

modulus-adapting via thiol−ene crosslinking reaction. Each of these parameters alters the modulus of the gel, indicating that the forming material responds to a range of electric input parameters. Taken together, we conclude that by manipulating an electric field, we provide a new mechanism for self-strengthening and an example of using electro-chemo-mechanical interplay to create an adaptive modulus response.

Since thermal rise was unmeasured in our bulk reaction, we wanted to perform a more careful analysis to understand how each element contributed to the thermodynamics of the process. To achieve this, we employed real-time electrorheology using a discovery hybrid rheometer (DHR) equipped with a dielectric accessory in shear mode at 20 °C (Fig. 3a). A volume of 0.25 mL of the same formulation of thiol−ene pre-reaction solution was placed on the rheometer plates and the plates were placed at a 0.5 mm gap distance. With the shortened gap distance, we calculated that the field strength would be increased four times compared to the bulk experiment, so we conducted these experiments at 2 $V_{rms}$. We observed that the material had a surprisingly rapid onset time (600 s) at 2 $V_{rms}$, 2 kHz, while the control group lacking ZnO remained in a liquid state (Fig. 3b). Next, we measured the storage modulus and the temperature of the material during the experiment. Our goal was to test if the onset of voltage led to increased heating and, therefore, either accelerated reactivity or triggered it. As in the previous experiment, the experimental group showed an increase in storage modulus from 0.02 to 179 kPa but at an onset time of 270 s (Fig. 3c). The mechanical analysis of the E-thiol−ene sample, as determined by strain−stress behavior and frequency-independent moduli, confirmed the formation of a robust gel after only 600 s (Supplementary Fig. 19). The control group again did not show an increase of modulus. More notably, coincident with the faster onset time, the ZnO showed an increase in temperature (0.5 °C over the control) as the cross-linking reaction released heat (Fig. 3d). From this, we concluded that at very high voltages, there is very modest heating that results from an applied electric field, but it is not enough to greatly affect reaction kinetics. The majority of the increase in temperature originated from the exothermic nature of the rapid onset of the reaction.

## Programmable multiple-modulus gel

After observing that the mechanical properties could be controlled by an electric field, we postulated that the precise placement of varied electric fields could be used to "program" the spatial stiffness of a single bulk material. To demonstrate this concept, we designed a parallel PTFE plate mold with adjustable ends to accommodate larger volumes and the placement of multiple electrodes. Multiple electrodes were placed along a parallel axis allowing addressable AC voltage in different configurations at these points in the sample (Fig. 4a). Based on our earlier experiments, three pairs of electrodes were addressed with configurable voltages (4, 12, and 8 $V_{rms}$ from left to right at 500 Hz) into the pre-casted organo-gel mixture composed of 7.8 wt% ZnO, 20 mmol TTT, 30 mmol EDT and 0.1 M $Et_4NBF_4$ in 15 mL DMF.

To model the changes in the material modules, we employed finite-element analysis (FEA; Fig. 4b)—assuming a linear relationship between the electric field strength and the applied voltage observed in our results (Fig. 2). The model showed that neighboring electric fields had little influence on each other. In the simulation, the direction of the electric field is neglected, as direction should not affect the randomly oriented nanoparticles in the material. Using the model to guide the placement, the electrodes were located on three distinct regions with different electric field strengths. Upon triggering the electric field in specific locations for 5 h, the thiol−ene crosslinking was programmed to occur at the desired speed to achieve regions of controlled stiffness in one bulk gel (Fig. 4c). For testing, the whole gel was divided into nine parts with a 13 mm × 17 mm cutter, among which the embedded part (marked region) between two paired electrodes had the highest

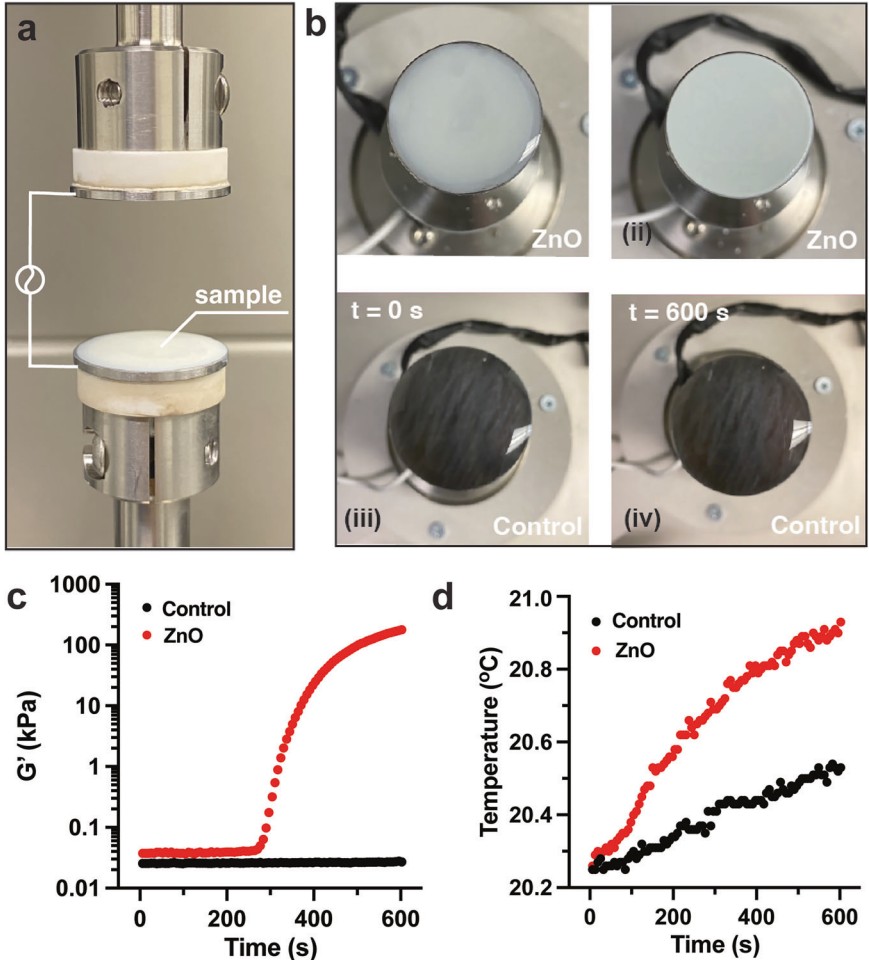

**Fig. 3 | Testing the in situ dynamic modulation of the mechanical properties under an electric field. a** The illustration of the setup of electric-field-adaptive properties testing inside a rheometer using oscillatory deformation. **b** Snapshots of testing samples on a rheometer. Images of the gelation sample (with ZnO) were captured before (i) and after (ii) applying a 2 $V_{rms}$ voltage. Images of the control sample (without ZnO) were captured before (iii) and after (iv) applying a 2 $V_{rms}$ voltage. **c** The time sweeps of storage modulus ($G'$) at 20 °C. **d** The change of temperature as a function of time.

modulus compared with the two areas adjacent to the focus of the electric field.

With this setup, we determined the storage modulus distribution of the thiol–ene gel along with the perpendicular direction of applied electric fields (Fig. 4d). The obtained modulus in each part that experienced a programmed voltage is consistent with the electric field distribution from the model. This experiment indicates that stiffness modulation is locally controlled within sections of a material, providing flexibility and efficiency to a wide variety of applications.

**Electro-adhesive based on thiol–ene crosslinking**
In the real-time electrorheology testing, we noted that the detachable plates remained adhered after polymerization from the thiol–ene crosslinking reaction. These adhesions were enhanced at shorter gap distances, higher voltage and longer time (0.1 mm, 2 $V_{rms}$, 15 min) (Supplementary Fig. 20). Thiol–enes and many radical-based reactions are used as commercial and industrial adhesives. We hypothesized that this electric field-induced adhesion between the dielectric material and the conductive plate was related to rheological behavior triggered by electric voltages. To date, there is only one known electro-adhesive, which uses remarkable diazrine chemistry[36]. However, this method has limited compatibility with current adhesives, including both thiol–ene and radical-mediated adhesives. To test our adhesive system properly,

we placed a thin layer of the same composition of thiol–ene reactant solution (0.025 mL) between two pieces of indium tin oxide (ITO)-coated glass plates (Fig. 5a). After applying the AC (500 Hz, 8 $V_{rms}$) for 5 min, we used lap shear testing to assess shear strength as a measure of the performance of the electro-adhesive. The dimensions of the adhesive layer were 25 mm × 25 mm × 0.03 mm. After 24 h, the adhesive sustained 260 N of shear force from 25 to 60 °C before loss of integrity (Fig. 5b). The adhesive strength of control samples without an electric field or ZnO nanoparticles was also evaluated. As depicted in Fig. 5c, the lap shear strength of the electro-adhesive surpasses that of control samples lacking either ZnO or an electric field, indicating strong adhesion occurred by the formation of electric field-triggered thiol–ene crosslinking reaction via ZnO nanoparticles. With the presence of both ZnO and E-field, the strength reaches 389.8 ± 42.0 kPa, which is comparable with commercial cyanoacrylate and epoxy adhesives (Supplementary Fig. 21), and higher than previously reported electro-adhesive (25–82 kPa) based on diazrine chemistry[36–38]. The adhesive strength is consistent with that of the monomers cured via traditional means[39]. We note that this is only a demonstration experiment with no optimization of properties, yet it is already comparable to some commercial products. When no voltage was applied between the electrodes, the ITO layers freely slid past one another, and the shear strength was dictated by the surface tension. Upon realizing that the thin film afforded very high uniform electric fields, we considered

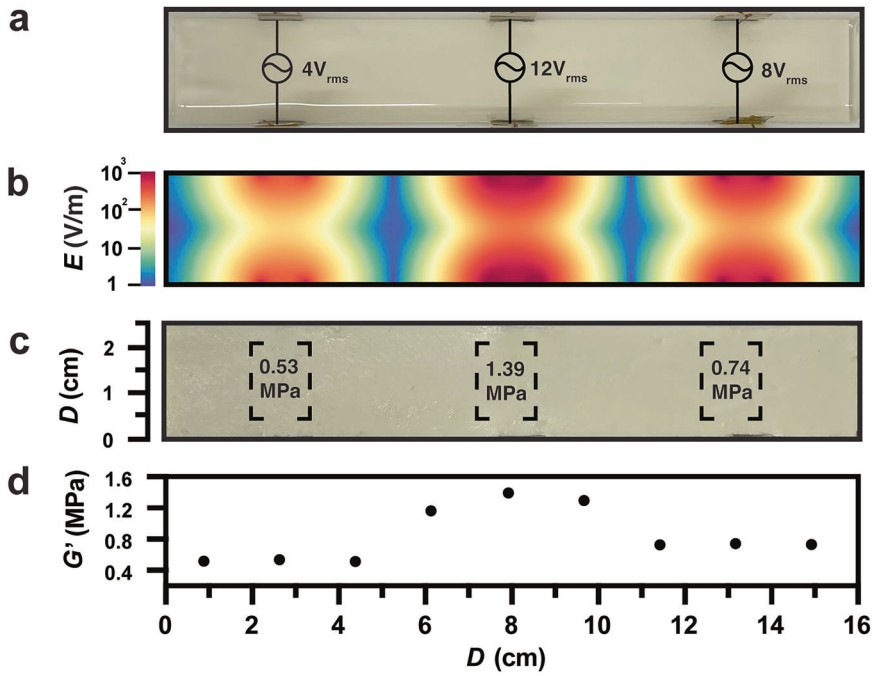

**Fig. 4 | Programming material stiffness through different AC inputs.**
**a** Illustration of the setup for a multiple-modulus gel with 3 pair electrode configuration by various voltages (4, 12, and 8 $V_{rms}$ from left to right at 500 Hz). **b** Finite-element simulation (via Python) of the electric field distribution within the sample. **c** Photograph of multi-stiffness organogel obtained after AC voltage application. **d** Storage modulus of electric field-induced thiol–ene organogel as a function of distance.

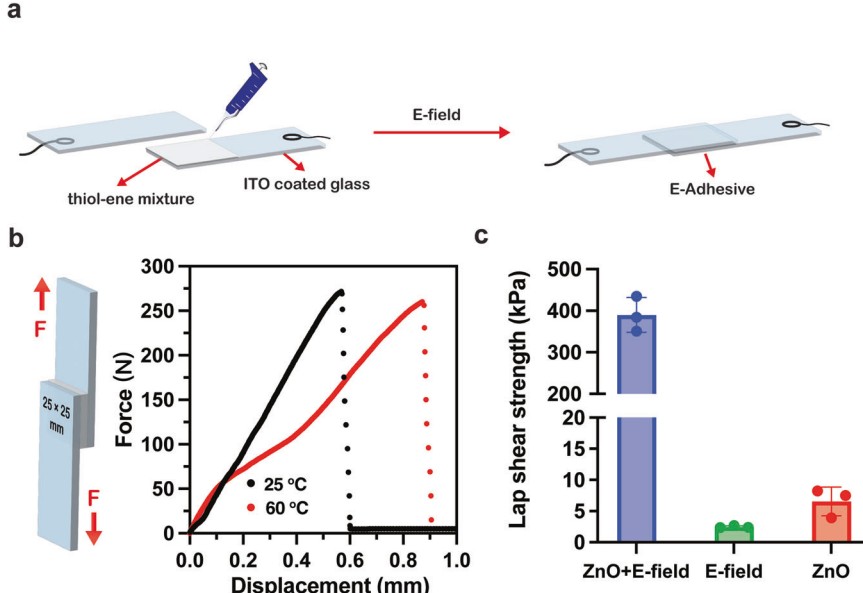

**Fig. 5 | Development of E-curing glue by piezo-chemistry. a** Schematic representation of the electro-adhesive setup used for adhesion testing and in situ triggering. **b** Graphical representation of lap shear adhesion test for electro-adhesive (left) and force–displacement curves for electro-adhesive after 24 h under 25 and 60 °C (right). **c** Lap shear strength of electro-adhesive and control samples on ITO-coated glass substrates at 25 °C.

that perhaps even modest DC fields might work for this specific application (Supplementary Movie 2). The ITO glass electrodes were connected to an AA battery (3 V, DC) as the electric energy source. After 5 min of 3 V bias, the reaction yielded an adhesive that bound the two ITO glass plates together. In contrast, the plates easily separated, suggesting that there was no adhesion in the absence of an electric trigger (Supplementary Movie 1).

In summary, we demonstrated what is, to our knowledge, one of the first methods to remotely induce a chemical reaction via an electric field. Using an electric field, in contrast to other methods of external control, allowed polymerization inside materials with individual segments that could be remotely tailored, programmed, and reconfigured. The electric field-driven reaction, in contrast and similarity to our previous work on mechanically mediated polymerization,

responded to electrical inputs of frequency, voltage, and exposure time. Using this method, we demonstrated the fabrication of an electric-field-adaptive multi-stiffness gel and electro-adhesive. We foresee applications for adaptive damping and variable-rigidity materials, adhesives, soft robotics, and potentially tissue engineering.

## Methods

### Material

Tri(ethylene glycol) divinyl ether (TEGDE), 2,2'-(ethylenedioxy)diethanethiol (EDT), 1,3,5-triallyl-1,3,5-triazine-2,4,6(1H,3H,5H)-trione (TTT), 5-(1,2-dithiolan-3-yl)pentanamide (DPA), methyl methacrylate (MMA), tetraethylammonium tetrafluoroborate (Et$_4$NBF$_4$), potassium iodide (KI), 4-$p$-methoxyphenol (MEHQ), zinc tetrafluoroborate hydrate, trifluoroacetic acid (TFA, 99%), indium tin oxide (ITO)-coated glass slide (70-00 Ω/sq, rectangular) were obtained from Sigma-Aldrich. Chloroform-d (CDCl$_3$, 99.8%) and 1,3,5-trimethoxybenzene (99.96%) TraceCERT® standards for quantitative NMR were purchased from Sigma-Aldrich. All chemicals were used as received. Dry DMF was obtained after purification using an in-house solvent purification system. ZnO (18 nm, 99.95%), BaTiO$_3$ (200 nm, 99.9%, tetragonal), ZnO–silane-coated (18 nm, 99.95%), and TiO$_2$ (100 nm, rutile 99.9+%) were purchased from US Research Nanomaterials Inc. Lead zirconate titanate (PZT, 100–150 nm, 99.9%) and Bismuth Ferrite (BiFeO$_3$, 80–100 nm, 99.9%) were purchased from Nanoshel LLC. Polypropylene cuboid vials (1.70 cm × 1.35 cm × 3.78 cm) were purchased from United States Plastic Corp. Platinum foils (0.1 mm thick, 99.99%) were purchased from Thermo Fisher Scientific. Gorilla super glue cyanoacrylate gel was purchased from Amazon.com, Inc. and Hardman double/bubble epoxy glue was obtained from Ellsworth Adhesive.

### Measurements

$^1$H nuclear magnetic resonance (NMR) spectra were recorded on a Bruker AVANCE II+ (500 MHz) spectrometer at 25 °C and processed them in MestReNova 14.2.0.

Gel permeation chromatography (GPC) was conducted in a Shimadzu Prominence LC instrument equipped with a Shimadzu Prominence LC-one Agilent PLgel 5 μm MiniMix-D separation column, eluent: stabilized THF (BHT 250 ppm), flow rate: 1 mL min$^{-1}$, temperature $T$ = 25 °C, a Shimadzu UV–VIS detector, a Wyatt DAWN HELEOS II multi-angle light scattering detector (658 nm laser), a Wyatt ViscoStar III detector and a Wyatt OptiLab T-rEX RI detector. The GPC system was calibrated with polystyrene (PS) standards.

Zeta potential measurement was conducted via a Mobius Zeta Potential Analyzer at the sample concentration of 0.2 mg mL$^{-1}$ in DMF under DC voltage from 5 to 50 V.

X-ray photoelectron spectroscopy (NEXSA G2 Keck-II XPS with monochromatic Al Kα X-ray radiation, emission current of 15 mA and hybrid lens mode, Manchester, UK) was used for the analysis of the surface of nanoparticles. Wide and narrow spectra were measured with a pass energy of 80 and 20 eV, respectively. XPS spectra were analyzed using Avantage software version 6.6.0 Beta. All spectra were calibrated using C 1$s$ peaks with a fixed value of 284.8 eV.

Mechanical characterization was conducted in a TA Instruments RSA-G2 dynamic mechanical analyzer (DMA) with 25 mm parallel steel plates. The sample was cut and mounted as a cuboid (2.5 mm thick) and compressed up to 0.1 N of force to achieve proper contact between the plates. Preliminary tests were performed to ensure that the applied oscillatory strain and the corresponding stress remained in the linear viscoelastic region over the whole temperature range. The frequency sweep experiments were performed at a frequency range from 0.1 to 10 Hz. The compression experiments were conducted at a constant rate of 0.01 mm/s.

Dynamic rheological experiments were performed using a Discovery Hybrid Rheometer (DHR-30), TA Instruments. The mixture of

TTT, EDT and supporting electrolyte in DMF was transferred through a pipette to the rheometer. A parallel-plate fixture (25 mm) with a solvent trap was utilized for rheological investigations. The evolution of gel formation as a function of time was captured using small-amplitude oscillatory shear experiments at a strain amplitude of 1% and a frequency of 1 Hz. The gap size was set to 0.5 mm. The experiments were performed at least three times, and representative results were displayed.

Lap shear tests were conducted on the material tester ZwickRoell zwickiLine Z0.5 (500 N loading cell) at 25 and 60 °C. The ITO-coated slides with thiol–ene adhesive, control and commercial adhesive samples (25 mm × 25 mm × 0.03 mm) were prepared and loaded with a tension rate of 10 mm/min.

Cyclic voltammetry (CV) was performed at 25 °C on a Gamry Reference 3000 potentiostat from Gamry Instruments. A platinum disc electrode, platinum wire and Ag/AgCl containing electrode were used as the working, counter and reference electrode, respectively. A typical cycle started from a negative potential; the cycle continues by sweeping the potential between +1.2 to −1.0 V with a scan rate of 0.2 V s$^{-1}$.

Vibration measurements were conducted at room temperature using a Polytec PSV 400 laser vibrometer. MATLAB R2022b was used to send a 10 V peak-to-peak ($V_{pp}$) sine-modulated Gaussian pulse (2000 Hz) to a Tektronix AFG3022C function generator, which was subsequently amplified by a PiezoDrive PD200 amplifier at 20 V/V. The amplified signal (200 $V_{pp}$) was then applied to the thiol–ene samples with copper tape electrodes. The PSV laser vibrometer records the velocity of the resulting vibration at a sampling frequency of 51.2 kHz, with the result being averaged 5000 times.

### Electric field-induced thiol–ene linear polymerization

In a typical experiment, ZnO (210 mg, 2.58 mmol) was dispersed in 1.5 mL DMF by ultrasound for 20 s and allowed to rest for 30 min. Two thiol–ene monomers, tri(ethylene glycol) divinyl ether (3 mmol, 606 mg, 0.612 mL) and 2,2'-(ethylenedioxy)diethanethiol (3 mmol, 546 mg, 0.489 mL) and Et$_4$NBF$_4$ (0.15 mmol, 32.5 mg) were added to the reaction mixture and took 2 mL mixed solution into a cuboid polypropylene vial. Finally, the plastic vial was typically attached to the power sources (AC or DC power) with two wired Pt electrodes inside. The applied voltages were typically operated at 500 Hz, 8 $V_{rms}$ for generating an alternating electric field unless otherwise noted. Aliquots were collected at intervals and analyzed using $^1$H-NMR for calculating the conversion.

### Electric field- induced thiol–ene crosslinked gelation

In a typical experiment, 210 mg of ZnO was dispersed in 1.5 mL DMF by ultrasound for 20 s and allowed to rest for 30 min. Two thiol–ene monomers, 1,3,5-triallyl-1,3,5-triazine-2,4,6(1H,3H,5H)-trione (2 mmol, 499 mg, 0.574 mL) and 2,2'-(ethylenedioxy)diethanethiol (3 mmol, 546 mg, 0.489 mL) and Et$_4$NBF$_4$ (0.15 mmol, 32.5 mg) were added to the reaction mixture in a polypropylene vial. Finally, the plastic vial was directly attached to the power sources (AC or DC power supply) with two wired Pt electrodes. A function generator was used to match the efficacy as an AC power supply, where the voltage amplitude was configured at the same level (4 $V_{rms}$) to allow efficiency comparisons.

### Electric field-induced thiol-acrylate linear polymerization

Electric field-induced acrylate-based polymerization was tested with a similar protocol to E-thiol–ene chemistry. Briefly, 200 mg of ZnO were dispersed in 1 mL DMF by ultrasound for 20 s and allowed to rest for 30 min. Then Et$_4$NBF$_4$ (0.23 mmol, 65 mg), EDT (0.61 mmol, 112 mg, 0.100 mL), and methyl methacrylate (MMA, 1 mL) were dissolved in the ZnO dispersion consecutively. 2 mL of this mixture was transferred into a plastic vial equipped with two Pt electrodes outside. Eventually, the sample was connected to an AC power supply (50 $V_{rms}$, 500 Hz) for 48 h.

## Electric field-induced disulfide polymerization

E-field-controlled disulfide polymerization was tested with a modified protocol described in the literature[31]. Briefly, 200 mg of ZnO were dispersed in 3 mL DMF first through ultrasonication. Then Et$_4$NBF$_4$ (0.3 mmol, 65 mg), 5-(1,2-dithiolan-3-yl) pentanamide (1 mmol, 205 mg), and EDT (12 μmol, 2.24 mg, 0.002 mL) were dissolved in the ZnO dispersion consecutively. In the end, the mixture was transferred into a plastic vial equipped with two Pt electrodes and connected with an AC power supply (8 V$_{rms}$, 500 Hz) for 3 h.

## Effect of ZnO concentration on thiol−ene crosslinked gelation

In a typical experiment, various concentrations of ZnO (1.50, 3.75, 5.63, 7.50 and 11.25 wt%) were dispersed in 1.5 mL DMF by ultrasound for 20 s and allowed to rest for 30 min. Two thiol−ene crosslinking monomers, TTT (2 mmol, 499 mg, 0.574 mL) and EDT (3 mmol, 546 mg, 0.489 mL) and Et$_4$NBF$_4$ (0.15 mmol, 32.5 mg) were added to the reaction mixture in a polypropylene vial. Then, the reaction mixture was divided into two batches: one batch was connected to an AC power source with two Pt electrodes and subjected to 500 Hz, 16 V$_{rms}$ for 3 h, while the other batch was transferred to an oven and cured at 100 °C for 24 h until fully crosslinked.

## Evaluation of EDT adsorption on nanoparticle surface

In a typical experiment, ZnO (210 mg, 2.58 mmol) or BaTiO$_3$ (210 mg, 2.58 mmol) was dispersed in 1.5 mL DMF by ultrasound for 20 s and allowed to rest for 30 min. 3 mmol EDT (546 mg, 0.489 mL) was then added to the suspension in a 50 mL centrifuge tube, and the mixture was vortexed for 3 h at 500 rpm. The resulting suspension was centrifuged at 4000×$g$ for 15 min to drive the nanoparticles to sediment. The supernatant solution was withdrawn using a transfer pipette, enough ethanol was added to bring the total volume of the suspension back to 50 mL, and the mixture was vigorously shaken and centrifuged again. This process was repeated so that a total of three centrifuge steps were performed. Finally, the washed nanoparticles were dried in a vacuum oven at 60 °C overnight.

## Electric field-induced thiol−ene reaction using thiol−ZnO nanoparticles

To further explore the surface chemistry, the dried thiol−ZnO from the previous experiments was dispersed in 1.5 mL DMF by ultrasound for 20 s and allowed to rest for 30 min. Tri(ethylene glycol) divinyl ether (3 mmol, 606 mg, 0.612 mL) and Et$_4$NBF$_4$ (0.15 mmol, 32.5 mg) were added to the reaction mixture and took 2 mL mixed solution was taken into a cuboid polypropylene vial. The plastic vial was connected to an AC power source with two-wired Pt electrodes inside, operating at 500 Hz and 16 V$_{rms}$ for 3 h. Control experiments with the same formulation but without applying an AC voltage were conducted to examine the effect of the electric field on the polymerization process. Aliquots were collected and analyzed using $^1$H-NMR for calculating the conversion.

## Evaluation of E-thiol−ene radical generation

In a typical experiment, ZnO (210 mg, 2.58 mmol) was dispersed in 1.5 mL DMF by ultrasound for 20 s and allowed to rest for 30 min. TEGDE (3 mmol, 606 mg, 0.612 mL), EDT (3 mmol, 546 mg, 0.489 mL), MEHQ (0.15−0.60 mmol) and Et$_4$NBF$_4$ (0.15 mmol, 32.5 mg) were added to the reaction mixture and underwent three cycles of freeze−pump−thaw treatment to eliminate dissolved oxygen. Finally, the plastic vial with 2 mL mixed solution was directly attached to the power sources (AC, 4 V$_{rms}$, 500 Hz) with two wired Pt electrodes. Aliquots were collected and analyzed using $^1$H-NMR for calculating the conversion.

## Evaluation of E-thiol−ene linear polymerization in glove-box

In a glove-box equipped with an AC power supply, ZnO (210 mg, 2.58 mmol) was weighed and mixed with 1.5 mL DMF in a sealed tube. The resulting ZnO dispersion was then transferred outside and

dispersed by ultrasound (sealed tube with Ar) for 20 s and allowed to rest for 30 min in the glove-box. Subsequently, TEGDE (3 mmol, 606 mg, 0.612 mL), EDT (3 mmol, 546 mg, 0.489 mL) and Et$_4$NBF$_4$ (0.15 mmol, 32.5 mg) were added to the reaction mixture and transferred outside the glove-box under Ar protection for three cycles of freeze−pump−thaw treatment to eliminate dissolved oxygen. Finally, the degassed reaction mixture was transferred back to the glove-box, and 2 mL mixed solution was dispensed into a plastic vial with two wired Pt electrodes inside. Finally, the plastic vial was then directly attached to the AC power sources. The applied voltages were operated at 500 Hz, 8 V$_{rms}$ for generating an alternating electric field at 1, 5, 15 and 30 min. Aliquots were collected at intervals and analyzed using $^1$H-NMR for calculating the conversion.

## Quantitative evaluation of EDT adsorption on ZnO nanoparticle surface

In a typical experiment, ZnO (210 mg, 2.58 mmol) and Et$_4$NBF$_4$ (0.15 mmol, 32.5 mg) were dispersed in 1.5 mL DMF by ultrasound for 20 s and allowed to rest for 30 min. 3 mmol EDT (546 mg, 0.489 mL) was then added to the suspension and 2 mL mixed solution was taken into a cuboid polypropylene vial. Then, the plastic vial was typically attached to the power sources (AC power supply) with two wired Pt electrodes inside. The applied voltages were operated at 500 Hz, 16 V$_{rms}$ for 3 h to generate an alternating electric field. Aliquots were collected and analyzed using $^1$H-NMR for calculating the EDT adsorption.

To further evaluate the amount of binding thiol onto the surface, the resulting suspension was transferred to a 50 mL centrifuge tube and centrifuged at 4000×$g$ for 15 min to drive the nanoparticles to sediment. The supernatant solution was withdrawn using a transfer pipette, enough ethanol was added to bring the total volume of the suspension back to 50 mL, and the mixture was vigorously shaken and centrifuged again. This process was repeated so that three centrifuge steps were performed. The washed nanoparticles (thiol−ZnO) were then dried in a vacuum oven at 60 °C overnight. Finally, the dried particles were weighed and dissolved using trifluoroacetic acid (TFA), and $^1$H-NMR was used to analyze the resulting solution to determine the amount of EDT with a calibration curve to ensure accurate quantification.

## Preparation of programmable multiple-modulus gel

Samples for the experiment were prepared by adding 2100 mg of ZnO to 15 mL DMF by ultrasound for 20 s and allowed to rest for 30 min. Then, TTT (20 mmol, 4990 mg, 5.74 mL), EDT (30 mmol, 5460 mg, 4.89 mL), and Et$_4$NBF$_4$ (1.5 mmol, 325 mg) were added to the reaction mixture in a U-shaped PTFE tunnel sealed with two slides at ends. In this case, multiple electric fields were achieved by fixing the position of parallel electrodes separately while delivering AC voltage in different configurations across the sample. Three pairs of electrodes with configured voltages (4, 12, and 8 V$_{rms}$ from left to right at 500 Hz) at different positions.

## Preparation of electro-adhesive

In a typical experiment, 210 mg of ZnO was dispersed in 0.5 mL DMF by ultrasound for 20 s and allowed to rest for 30 min. Two thiol−ene monomers, TTT (2 mmol, 499 mg, 0.574 mL) and EDT (3 mmol, 546 mg, 0.489 mL) and Et$_4$NBF$_4$ (0.05 mmol, 10.8 mg) were added to the reaction mixture in a polypropylene vial. Finally, 30 μL of the mixture was directly applied to the ITO-coated glass, which was wired with the power sources (AC power supply). A series of 2 AA batteries (3 V) was used to supply as a DC power source.

## Electric field simulation with Python

The electric field within the thiol−ene solution is numerically simulated using Python, based on Jacobi relaxation. With the electric potential of

the electrodes and far fields fixed, the potential field is iteratively updated based on the surrounding fields from the last iteration. The solution to the electrostatic potential Laplace's equation is achieved after thousands of iterations. The thiol–ene solution was simulated as two-dimensional because it is homogeneous along the vertical axis. In the simulation, six electrodes are placed on the long edges of the solution, with fixed voltages that are the maximum voltage from the applied AC power. Relaxation time of the electric field is at the scale of nano-seconds, which is much faster than the period of the AC power (0.5 ms), so that a simulation with fixed voltage represents a moment of the AC circuit, and all the other moments are only different by the amplitude. The simulated solution follows the shape and size of the experimental settings strictly. A grid of 0.1 mm is set to achieve high resolution. First, the electric potential for each grid point is simulated with Jacobi relaxation, and then the electric field is calculated based on the potential difference between neighboring positions. The code of this simulation can be found at https://github.com/imechaozhang/Thiol-ene-electric-field-simulaiton. Only the magnitude of the electric fields is shown in Fig. 4c because the direction of the electric field does not make a difference for the nanoparticles in the solution. In the logarithm plot in Fig. 4c, the electric field is incremented by 1 V/m in order to take the logarithm values.

## Electric field-induced vibration via vibrometer

To evaluate the magnitude of electrically stimulated vibration, two thiol–ene samples were fabricated using TTT (20 mmol, 4490 mg, 5.74 mL), EDT (30 mmol, 5460 mg, 4.89 mL), DMF (4.5 mL), and 5850 mg of nanoparticles (ZnO for the test sample, $TiO_2$ for the control), and then cured in a UV oven at 40 °C for 24 h. The ZnO sample was pre-treated with a 60 $V_{pp}$ signal at 500 Hz for 3 h. The cross-sectional area of the resulting samples was 1.27 cm × 1.27 cm. The ZnO and control sample thicknesses were 1.80 and 1.70 cm, respectively. The resulting time domain and frequency domain of the displacement (calculated by dividing the measured velocity signal by the wavepacket center angular frequency) are shown in Supplementary Fig. 18a, b.

## Data availability

All data that support the findings and results of this study are available in the paper and its Supplementary Information. Additional data is available from the corresponding author upon request.

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

## Acknowledgements

We thank Dr. Philip Griffin for the helpful discussion on the DHR measurements. Parts of this work were carried out at the Soft Matter Characterization Facility, NMR Facility and the Materials Research Science and Engineering Center of the University of Chicago and the Keck-II facility of Northwestern University's NUANCE Center. The work was supported by AFOSR COE 5-29168, NSF CHE-1710116, ARO W911NF-17-1-0598 (71524-CH) and MRSEC NSF DMR-2011854. I.F. acknowledges support from the NDSEG Fellowship Program through the US Army Research Office. N.B. acknowledges support from the US Army Research Office (Grant No. W911NF-20-2-0182).

## Author contributions

J.W., Z.W., J.A., and A.E.K. conceived the concept. J.W., Z.W., and A.E.K. designed the experiments. Z.W., J.W., J.A., I.F., C.H., K.Q., K.K., Y.D., P.H., N.B., C.L., and M.M. performed the experiments and analyzed the data. C.Z. conducted the Finite Element Simulation. All authors participated in writing the manuscript.

## Competing interests

The authors declare no competing interests.
