## [Peer Review File · Nature Communications]

Programmable Material via Thiol-ene Polymerization Initiated by Electric-field Induced Thiyl Radical on Piezoelectric ZnO

Corresponding Author: Professor Aaron Esser-Kahn

Version 0:

Reviewer comments:

Reviewer #1

(Remarks to the Author)

[Note from the Editor: Reviewer #1 was asked to look also over the response given to Reviewer #4]

This manuscript presents an innovative approach to inducing thiol-ene polymerization using electric fields. The central advance lies in the use of piezoelectric zinc oxide (ZnO) nanoparticles as transducers that convert electric fields into localized mechanical strain. This strain, in turn, initiates radical-mediated thiol-ene reactions without the need for direct electrode contact and enabling spatial control over the reaction. The authors convincingly demonstrate this capability by generating spatially defined multi-modulus gels, supported by both finite element modeling and experimental validation.

The manuscript was previously considered for publication, and as noted by prior reviewers, the work contains several creative and compelling aspects. The authors have suggested a reasonable mechanism of thiyl radical generation upon piezoelectric stimulation of ZnO nanoparticles, providing a promising foundation for further development of electrically controlled thiol-ene polymerizations and other chemistries that rely on thiyl radical initiation.

Overall the work is well done, and the only major criticism is how the work is represented and authors' claims. However, it would be relatively straightforward for the authors to revise their manuscript, as suggested below, to be suitable for publication.

--

Specific feedback and suggested revisions:

There appears to be a miscommunication between Reviewer 1 and the authors regarding the manuscript title. Reviewer 1 suggests revising the phrase "Programmable Materials Stiffness," noting that an increase in stiffness is a general consequence of polymerization due to the formation of higher molecular weight species. From this perspective, most photopolymerizations could be described as enabling "programmable stiffness." While this does not diminish the importance of achieving polymerization in optically opaque materials, the reviewer's concern is that the current title uses overly general or stylistic language in place of a more precise technical description.

Indeed, the title should more clearly reflect the specific scope and findings of the work. This study does not demonstrate a general piezoelectric nanoparticle-mediated effect, but rather a unique interaction between ZnO and thiol leading to thiyl radical formation and thiol-ene polymerization. A more accurate and informative title might be: "Piezoelectric ZnO-Induced Thiyl Radical Formation for Electric-Field Directed Thiol-Ene Polymerization"

Additionally, the authors include the statement:

"Critically, our E-thiol-ene polymerization kinetics is similar to those thiol-ene which are initiated by light – indicating that this process is of similar efficiency and rate as light-based approaches of similar power."

This assertion is somewhat ambiguous, particularly in its reference to "similar power." If the authors are drawing a

quantitative comparison between electric field power input and light-based initiation, it would be important to provide the underlying calculations and assumptions (ideally in the Supplementary Information) for transparency and reproducibility.

In the concluding claim,

“In summary, we demonstrated what is, to our knowledge, one of the first methods to remotely induce a chemical reaction via an electric field,”

is overly broad and should be revised or removed. The phrase “one of the first” requires qualification: What prior methods exist, how do they compare, and how many are your methods referring to? Moreover, electric fields have been used in various contexts—including hysteresis heating of ferromagnetic nanoparticles and redox reactions to induce chemical transformations. Without clarifying how the current approach differs fundamentally from these precedents, the statement misrepresents the novelty of the work.

Regarding Reviewer 4's Comment 1: It is not clear that the authors have fully addressed the question of what specifically makes ZnO unique. The authors note that the combination of chemical interactions with thiols and piezoelectric properties is the main justification. While this assertion has merit, it remains a hypothesis that ZnO is singular in this respect. For a communication-style manuscript, the somewhat open-ended nature of this claim—without demonstrating or even naming other potential piezoelectric catalysts—is acceptable. However, the authors absolutely need to revise the title to avoid implying that this is a broadly applicable phenomenon. Rather than stating “...Reaction via Piezo-Chemistry...”, the title should be changed to “...Reaction with ZnO...” since this is the only system that has been demonstrated.

Regarding Reviewer 4's Comment 2: The reviewer offered the insightful observation that thiols could act as ligands and, under certain conditions, dissolve ZnO, generating Zn ions – it should be noted that this could trigger acid-catalyzed thiol–ene reactions (e.g., Uchiyama et al., *Angew. Chem., Int. Ed.*, 2020). The authors conducted additional experiments to examine the possibility of Zn ion formation as the initiating mechanism. Their findings, which they report as supporting a radical-mediated pathway, appear reasonable. The authors have clarified this in their response, and I find their efforts to consider and rule out this alternative mechanism to be thorough within the scope of a communication.

Regarding Reviewer 4's Comment 3: Reviewer 4 argued that more extensive mechanistic studies would be necessary for a journal such as *Nature Materials*. While further work is needed to conclusively exclude all alternative pathways, the level of evidence is appropriate for a communication intended to report preliminary results. They have committed to clarifying in the manuscript that this is an initial demonstration rather than a definitive mechanistic study. I consider this a fair justification given the format and scope of the article.

Regarding Reviewer 4's Comment 4: Regarding the observation that polymerization continued for hours after the applied voltage was removed (Figure S10). The authors' explanation of difficulty of biomolecular termination is not likely the reason. Thiol–ene resins without inhibitors are well known to slowly polymerize over time. Therefore, it is not concerning that these materials continue to polymerize after the field is switched off.

Comment 5: The reviewer observes the clear differences between the kinetics of this reaction and the more traditional UV curing and further inquires about how ZnO affects the resulting material properties. While the authors argue that curing without light offers potential advantages for adhesive applications, it is important to note that the data in Figure S25 underscore significant limitations of this approach. Specifically, the polymer exhibits a modulus of less than 200 kPa at ZnO loadings below 5 wt%, indicating extremely weak mechanical performance likely due to low conversion. Figure S26 shows that higher modulus values (still in the rubbery range around ~1 MPa) can be achieved, but only after an impractical 24-hour post-cure at 100 °C. This requirement raises the question: if extended heating is necessary, why not simply use a conventional thermal radical initiator instead? In any case, it is still a unique initiation mechanism, which may merit its report in *Nature Communications*.

We are deeply grateful to the reviewers for their valuable suggestions, which have significantly improved the paper. Please find our point-to-point response below:

Point to point response

Reviewer #1:

This manuscript presents an innovative approach to inducing thiol-ene polymerization using electric fields. The central advance lies in the use of piezoelectric zinc oxide (ZnO) nanoparticles as transducers that convert electric fields into localized mechanical strain. This strain, in turn, initiates radical-mediated thiol-ene reactions without the need for direct electrode contact and enabling spatial control over the reaction. The authors convincingly demonstrate this capability by generating spatially defined multi-modulus gels, supported by both finite element modeling and experimental validation.

The manuscript was previously considered for publication in Nature Materials, and as noted by prior reviewers, the work contains several creative and compelling aspects. The authors have suggested a reasonable mechanism of thiyl radical generation upon piezoelectric stimulation of ZnO nanoparticles, providing a promising foundation for further development of electrically controlled thiol-ene polymerizations and other chemistries that rely on thiyl radical initiation.

Overall the work is well done, and the only major criticism is how the work is represented and authors' claims. However, it would be relatively straightforward for the authors to revise their manuscript, as suggested below, to be suitable for publication.

--

Specific feedback and suggested revisions:

There appears to be a miscommunication between Reviewer 1 and the authors regarding the manuscript title. Reviewer 1 suggests revising the phrase "Programmable Materials Stiffness," noting that an increase in stiffness is a general consequence of polymerization due to the formation of higher molecular weight species. From this perspective, most photopolymerizations could be described as enabling "programmable stiffness." While this does not diminish the importance of achieving polymerization in optically opaque materials, the reviewer's concern is that the current title uses overly general or stylistic language in place of a more precise technical description.

Indeed, the title should more clearly reflect the specific scope and findings of the work. This study does not demonstrate a general piezoelectric nanoparticle-mediated effect, but rather a unique interaction between ZnO and thiol leading to thiyl radical formation and thiol-ene polymerization. A more accurate and informative title might be: "Piezoelectric ZnO-Induced Thiyl Radical Formation for Electric-Field Directed Thiol-Ene Polymerization"

Re: We respectfully disagree with this point and the subsequent suggested title. While the current scope of this reaction is limited to this initial chemical pair, we have conducted extensive work demonstrating that it is mediated via a piezoelectric mechanism. We believe that this is a critical insight in our findings and should be included. We are willing to limit the remainder of the title to focus on the specific chemistry. Additionally, we would like to note that nearly every paper focusing on a novel material property driven by new chemistry uses a single underlying reaction. As this is the standard in the field, we feel it would be appropriate to consider that in our case as well. We programmed the material using multiple, identical pads to deliver different E-fields in a controlled, predictable fashion simultaneously, and in a predictive fashion – a feat that has not been achieved with light to our knowledge. We demonstrated that the modulus could be directly modeled and programmed by using an FEM to predict the modulus and then deliver it. If it is not clear, we can change the order of that figure. While it could potentially be done with light, it has not been realized to date. If such a method exists, we would be interested in references or examples. We have cited all relevant sources and papers with similar effects, and if the reviewers are aware of alternative examples, we greatly appreciate their input.

To better reflect the scope of the publication and the nature of the our findings, we changed the title to "**Programmable Material via Thiol-Ene Polymerization Initiated by Electric-field Induced Thiyl Radical on Piezoelectric ZnO**"

Additionally, the authors include the statement:

"Critically, our E-thiol-ene polymerization kinetics is similar to those thiol-ene which are initiated by light – indicating that this process is of similar efficiency and rate as light-based approaches of similar power."

This assertion is somewhat ambiguous, particularly in its reference to "similar power." If the authors are drawing a quantitative comparison between electric field power input and light-based initiation, it would be important to provide the underlying calculations and assumptions (ideally in the Supplementary Information) for transparency and reproducibility.

Re: Thanks for the insightful suggestion. We have now provided a detailed set of calculations for this in the SI (Supplementary Table 7). We also provided references that included all these power calculations. We apologize for not making them easier to locate. We have summarized them into a single, clear table for further consideration. We appreciate the reviewers' suggestions and believe they strengthen the paper.

Supplementary Table 7. Energy efficiency comparison between the e-field curing system (this work) and the existing light curing in the literature.

Entry	Power source	Power intensity	Curing time	Ref
1	AC generator	58 mW/cm ²	~10 min (without initiator)	This work
2	Uvitron UV 1080 Flood Curing System	120 mW/cm ²	<1min (with initiator)	4
3	EFOS Acticure (mercury vapor lamp)	15 mW/cm ²	15-30 min (without initiator)	5

In the concluding claim,

“In summary, we demonstrated what is, to our knowledge, one of the first methods to remotely induce a chemical reaction via an electric field,”

is overly broad and should be revised or removed. The phrase “one of the first” requires qualification: What prior methods exist, how do they compare, and how many are you referring to? Moreover, electric fields have been used in various contexts—including hysteresis heating of ferromagnetic nanoparticles and redox reactions to induce chemical transformations. Without clarifying how the current approach differs fundamentally from these precedents, the statement misrepresents the novelty of the work.

Re: Thanks for the comment. We addressed this point earlier in the paper, where we compared our work with existing systems.

“However, among all stimuli, an electric field uniquely excels at control, sensing, and distribution of power^{7,8}. Several examples in the literature show that an electric field induces physical changes in a material, such as phase transitions, which subsequently results in a change in shape or stiffness⁹. These materials are often referred to as electro-programmable materials, and possess responsive reversibility, large-scale variability, and multiple-input compatibility for displaying localized stiffness modulation⁸. For example, Silberstein et al. reported a comprehensive computational study on the stiffening of polyelectrolytes under an electric field¹⁰. The study showed that the elastic modulus of a polyelectrolyte can increase by 45% under a simulated electric field ($E = 0.2 \text{ V/m}$) due to better chain alignment and an increase in number of ionic crosslinks. However, the stiffening is physically induced and only occurs with charged polymers while under constant application of an electric field, meaning the material would soften once the electric field is removed¹⁰. In contrast, electrochemistry has been used to produce chemical reactions within materials resulting in permanent structural modifications^{1,8}. However, electrochemical reactivity is highly dependent on both the surface of the electrodes as well as the conductivity of the material¹¹. Chemical reactions that can be triggered and controlled through the use of electric fields with minimal current. Nevertheless, it has been difficult to use electric fields to directly induce chemical reactions in a material – relying only on surface electrodes or infeasibly high electric field (107 to 109 V/m) to mediate chemical changes.”

We appreciate the reviewer’s concern, and we apologize if this section is not clear, but we believe we have provided sufficient context and citations to show the distinctive nature of this new reaction. We demonstrated that heating a material with a massive E field is more inefficient than simply applying heat in the same manner. One can also do electrochemistry using electrodes, as we noted. We are demonstrating the first example where an E-field directly drives chemistry through a material by making and breaking chemical bonds. Only one precedent exists for this phenomenon (Chem. Mater. 2020, 32, 2440–2449), and our system offers significant improvements: 1) **Lower voltage requirement** (1–10 V vs. 40–80 V in literature). 2) **Faster curing speed** (<5 min vs. 10 min for existing volt-glue). 3) **Lower power consumption** ($\sim 58 \text{ mW/cm}^2$, calculated from DHR, vs. $\sim 200 \text{ mW/cm}^2$ in prior work).

Finally, as other reviewers noted, our system achieves efficiency comparable to light-based reactions.

In summary, we demonstrated what is, to our knowledge, the first high-efficiency, long-range, low-power method to directly make and break chemical bonds remotely modulated via an electric field.

Regarding Reviewer 4's Comment 1: It is not clear that the authors have fully addressed the question of what specifically makes ZnO unique. The authors note that the combination of chemical interactions with thiols and piezoelectric properties is the main

justification. While this assertion has merit, it remains a hypothesis that ZnO is singular in this respect. For a communication-style manuscript, the somewhat open-ended nature of this claim—without demonstrating or even naming other potential piezoelectric catalysts—is acceptable. However, the authors absolutely need to revise the title to avoid implying that this is a broadly applicable phenomenon. Rather than stating “...Reaction via Piezo-Chemistry...”, the title should be changed to “...Reaction with ZnO...” since this is the only system that has been demonstrated.

Re: Thanks for the suggestion. Now the title has been changed into “Programmable Material via Thiol-Ene Polymerization Initiated by Electric-field Induced Thiyl Radical on Piezoelectric ZnO”

Regarding Reviewer 4's Comment 2: The reviewer offered the insightful observation that thiols could act as ligands and, under certain conditions, dissolve ZnO, generating Zn ions – it should be noted that this could trigger acid-catalyzed thiol-ene reactions (e.g., Uchiyama et al., *Angew. Chem., Int. Ed.*, 2020). The authors conducted additional experiments to examine the possibility of Zn ion formation as the initiating mechanism. Their findings, which they report as supporting a radical-mediated pathway, appear reasonable. The authors have clarified this in their response, and I find their efforts to consider and rule out this alternative mechanism to be thorough within the scope of a communication.

We greatly appreciate the support of our scientific argument from this reviewer.

Regarding Reviewer 4's Comment 3: Reviewer 4 argued that more extensive mechanistic studies would be necessary for a journal such as *Nature Materials*. While further work is needed to conclusively exclude all alternative pathways, the level of evidence is appropriate for a communication intended to report preliminary results. They have committed to clarifying in the manuscript that this is an initial demonstration rather than a definitive mechanistic study. I consider this a fair justification given the format and scope of the article.

Re: Thank you very much for justifying our scope of study.

Regarding Reviewer 4's Comment 4: Regarding the observation that polymerization continued for hours after the applied voltage was removed (Figure S10). The authors' explanation of difficulty of biomolecular termination is not likely the reason. Thiol-ene resins without inhibitors are well known to slowly polymerize over time. Therefore, it is not concerning that these materials continue to polymerize after the field is switched off.

Re: We greatly appreciate this reviewer supporting our data.

Comment 5: The reviewer observes the clear differences between the kinetics of this reaction and the more traditional UV curing and further inquires about how ZnO affects the resulting material properties. While the authors argue that curing without light offers potential advantages for adhesive applications, it is important to note that the data in Figure S25 underscore significant limitations of this approach. Specifically, the polymer exhibits a modulus of less than 200 kPa at ZnO loadings below 5 wt%, indicating extremely weak mechanical performance likely due to low conversion. Figure S26 shows that higher modulus values (still in the rubbery range around ~1 MPa) can be achieved, but only after an impractical 24-hour post-cure at 100 °C. This requirement raises the question: if extended heating is necessary, why not simply use a conventional thermal radical initiator instead? In any case, it is still a unique initiation mechanism, which may merit its report in Nature Communications.

Re: Thank you for the feedback. We believe there is a misunderstanding on Figure S26. The comparison should be between Fig 25 and Fig 26. To clarify, our method provides a more power-efficient, faster method of forming a bonded material in that time range the traditional method. We wanted to clarify that the limit of this particular thiol-ene polymer was approximately 1 MPa. We never stated that our e-cured thiol-ene requires a post-curing process to reach a specific modulus. Instead, we were attempting to show that it has a similar modulus after 3 hrs as an equivalent thiol-ene that was purely cured by heat for 24 hrs at 100 °C. We do acknowledge that this requires a higher load of ~7% ZnO. This is a preliminary set of results and our goal was simply to show that it was possible to achieve equivalent strength to thermal curing, we openly acknowledge is an unoptimized formulation.

For further clarity, we have included Figure S27, which directly compares the modulus of e-field-cured and thermally cured thiol-ene. Since thermal curing is the conventional method, we normalized its modulus to 100% for each ZnO wt%. This allows us to quantitatively assess the degree of curing completion achieved via e-field curing.

Supplementary Figure 25. Storage modulus of the thiol-ene gel as a function of ZnO concentration (1.50 to 11.25 wt%) under AC voltage (16 Vrms, 500 Hz, 3 h). Measurements were performed at a fixed strain amplitude of 1%.

Supplementary Figure 26. Storage modulus of the thiol-ene gel as a function of ZnO concentration (1.50 to 11.25 wt%) under heat (100 °C, 24 h) until fully crosslinked. Measurements were performed at a fixed strain amplitude of 1%.

Supplementary Figure 27. Comparison of storage modulus (G') of thiol-ene gels formed by AC field-induced gelation (16 V_{rms} , 500 Hz, 3 h). and thermal curing (100 °C, 24 h) as a function of ZnO concentration (1.50 to 11.25 wt%). Measurements were performed at a fixed strain amplitude of 1%.